# EBV and Lymphomagenesis

**DOI:** 10.3390/cancers15072133

**Published:** 2023-04-04

**Authors:** Daniel G. Sausen, Ayeman Basith, Syed Muqeemuddin

**Affiliations:** 1School of Medicine, Eastern Virginia Medical School, Norfolk, VA 23507, USA; 2Department of Internal Medicine, Eastern Virginia Medical School, Norfolk, VA 23507, USA; 3Independent Researcher, Norfolk, VA 23510, USA

**Keywords:** EBV, latent proteins, cancer, lymphoma, treatment

## Abstract

**Simple Summary:**

Epstein–Barr virus is a highly prevalent virus associated with a multitude of diseases, including autoimmune conditions such as multiple sclerosis and many types of cancer. As such, it is imperative to have a foundational understanding of this virus. This review discusses the contribution of the Epstein–Barr virus to key hematologic malignancies with a focus on the roles of latent proteins, including diffuse large B-cell lymphoma, Hodgkin lymphoma, Burkitt lymphoma, NK/T-cell lymphoma, and primary CNS lymphoma. It then provides a brief overview of treatment for each of these diseases.

**Abstract:**

The clinical significance of Epstein–Barr virus (EBV) cannot be understated. Not only does it infect approximately 90% of the world’s population, but it is also associated with numerous pathologies. Diseases linked to this virus include hematologic malignancies such as diffuse large B-cell lymphoma, Hodgkin lymphoma, Burkitt lymphoma, primary CNS lymphoma, and NK/T-cell lymphoma, epithelial malignancies such as nasopharyngeal carcinoma and gastric cancer, autoimmune diseases such as multiple sclerosis, Graves’ disease, and lupus. While treatment for these disease states is ever evolving, much work remains to more fully elucidate the relationship between EBV, its associated disease states, and their treatments. This paper begins with an overview of EBV latency and latency-associated proteins. It will then review EBV’s contributions to select hematologic malignancies with a focus on the contribution of latent proteins as well as their associated management.

## 1. Introduction

Epstein–Barr virus (EBV) is a wide-spread viral pathogen afflicting approximately 90% of the world’s population [1]. Infection most typically occurs following exposure to infected oral secretions, although routes of transmission are recognized, including allograft transplantation and blood transfusion [1].

Primary infection acquired at a young age is often asymptomatic. Symptomatic primary infection usually occurs in adolescence or young adulthood and results in infectious mononucleosis [2], which presents with symptoms of fever, pharyngitis, tender adenopathy, and fatigue [2,3]. Interestingly, individuals with asymptomatic infection have demonstrated viral loads identical to those with symptomatic infections. The presence of symptoms has been found to be instead dictated by the host immune response [4].

A key feature of EBV infection is its ability to establish latent infections in B cells [5]. Long-term infection of B cells coupled with cellular transformation and viral protein expression are key elements in EBV-mediated carcinogenesis [6]. Given these abilities, it is not surprising that EBV is associated with a host of malignancies [6]. Amongst the hematologic malignancies, EBV is associated with diffuse large B-cell lymphoma (DLBCL) [7], classic Hodgkin lymphoma [8], NK/T-cell lymphoma [9], post-transplant lymphoproliferative disorders (PTLD) [10], Burkitt lymphoma [11], and primary CNS lymphoma [12]. EBV has also been linked to multiple epithelial malignancies [13], including gastric cancer [14] and nasopharyngeal cancer [15]. These EBV-associated malignancies were thought to be responsible for 239,700–357,900 new cases and 137,900–208,700 deaths in 2020 [16]. It is estimated that EBV causes 1.8% of all cancer-related deaths worldwide [17]. Indeed, just EBV-associated gastric cancer, nasopharyngeal carcinoma, Burkitt lymphoma, and Hodgkin lymphoma were responsible for 164,000 deaths [18]. In addition, there is less conclusive evidence linking EBV to the pathogenesis of multiple other epithelial malignancies [13]. EBV proteins, which will be discussed further in this article, are key to the oncogenic process [19].

Beyond malignancies, EBV has been linked with several other pathologic states, including encephalitis [20], oral hairy leukoplakia [21], EBV-associated post-transplant lymphoproliferative disorder [22], Alzheimer’s disease [23], and autoimmune conditions such as Graves’ disease [24], systemic lupus erythematosus, Sjögren’s syndrome, and rheumatoid arthritis [25]. In addition, it was recently demonstrated that EBV-infected patients have a 32-fold increased risk of developing multiple sclerosis [26].

EBV, also known as Human herpesvirus 4 (HHV-4), is a member of the Gammaherpesvirinae subfamily of Herpesviridae family. Its viral structure, which is similar to those of other herpesviruses, includes a double-stranded DNA core with a surrounding icosahedral capsid [27]. There is a surrounding envelope with glycoprotein spikes as well as a protein tegument between the capsule and the envelope [28]. Envelope glycoproteins have multiple important roles (reviewed in [29]) ranging from cellular entry [30] and viral assembly [31] to immune evasion [32]. EBV glycoproteins help dictate its preferential tropism for B cells, which has important implications in generating B-cell pathologies [33].

This review will begin with a brief overview of B-cell latent proteins and their role in B-cell transformation. This will be followed by a discussion of other proteins expressed by EBV that are important in the oncogenic process. We will then review the pathogenesis (with a focus on latent proteins) and treatment options for key hematologic malignancies associated with EBV.

## 2. EBV Latency

EBV can establish multiple types of latent infection, termed latency 0, I, II, and III, depending on factors such as cell type, local environment, and time since infection. They are differentiated from one another by the genes expressed [34]. Type 0 latency is typically seen during latent infection of memory B cells. No latent proteins are expressed, although Epstein–Barr virus-encoded small RNAs (EBERs) are seen. In type I latency, Epstein–Barr nuclear antigen (EBNA) 1 is expressed in addition to EBERs. Type II latency features the expression of EBERs, EBNA1, latent membrane protein (LMP) 1, and LMP2. Type III latency features the most expansive expression profile. The expressed proteins include EBNA2, EBNA3, and EBNA-leader protein (EBNA-LP) in addition to all those expressed in type II latency [35,36].

Malignancies are more strongly associated with specific latent states [37]. For example, Burkitt’s lymphoma is associated with type I latency while DLBCL is associated with type II latency and, less frequently, type III latency. PTLD is usually associated with type III latency [37] and type II latency is seen with Hodgkin lymphoma [38] as well as NK/T lymphomas [39]. Figure 1 demonstrates patterns of EBV latent gene expression.

### Brief Overview of EBV Latent Proteins

A primary role of EBNA1 is viral genome maintenance [40]. This process is reliant on EBNA1 forming a cross-link with the EBV origin of plasmid replication (oriP) [41]. Furthermore, it has been shown that EBNA1 contains SUMO-interacting motifs (SIMs) that are essential in maintaining latency through inhibition of the Small-Ubiquitin-related modifier (SUMO) 2 complex including STUB1, KAP1, and USP7 [42]. This protein has also been implicated in immortalization and survival. Indeed, it is required for efficient transformation [43]. Its expression has been associated with increased levels of reactive oxygen species (ROS) such as 8-Oxoguanine-glycosylase-1 (OGG1). Importantly, it also upregulates antioxidant pathways such as MTH1 pyrophosphatase. Furthermore, MTH1 inhibitors cause DNA damage in EBNA1^+^ cells, and MTH1 upregulation limits oxidative stress caused by EBV [44]. It also stimulates expression of other latent genes [40]. Recent evidence has indicated that EBNA1 has many roles beyond viral episome maintenance, including viral persistence, cell survival, and oncogenesis [40].

EBNA2 is expressed very early in the infection cycle [45]. Pich et al. found that it was the only essential latent gene for early latent gene reprogramming [46]. These results are consistent with previous experiments showing that EBNA2 is essential in B-cell transformation [47]. However, in vivo experiments indicate that EBNA2 deletion does not preclude the establishment of infection [48] or development of cancer [49]. Notably, EBNA2 expression is highly efficient; experiments have shown that a single detectable EBV genome is sufficient to drive EBNA2 expression [50].

EBNA2 also plays important roles in altering cellular protein expression profiles. For example, CD23, also called Blast2/Fc epsilon RII, is a B-cell activation antigen that is upregulated by EBNA2 [50,51]. In addition, Yanagi et al. used RNAseq to show that EBNA2 induces programmed death ligand 1 (PD-L1) expression to further evade host immunity [52]. Their findings regarding PD-L1 are consistent with previous results demonstrating increased PD-L1 expression in Burkitt lymphoma and DLBCL secondary to decreased miR-34a expression, which targets PD-L1. The postulated mechanism is as follows: EBNA2 recruits EBF1 to the miR-34a promoter with the subsequent downregulation of miR-34a expression and increased PD-L1 levels [53]. Other genes induced by EBNA2 include those involved in the cell cycle, metabolic processes, membrane morphogenesis, and vesicle regulation. Immune signaling genes have been shown to be suppressed by EBNA2 [52]. EBNA2 also plays a role in immune evasion through the downregulation of human leukocyte antigen (HLA) class II molecules by decreasing CIITA expression [54].

Like EBNA2, EBNA-LP is expressed very early in the infection cycle [45]. The primary role of EBNA-LP is to act as a co-activator of EBNA2 [55]. In conjunction with EBNA2, EBNA-LP is involved in B-cell transformation through activation of viral and cellular transcription [56], for example, through their effects on EBF1 and RBP-jκ [57]. There is some evidence supporting a broader role for EBNA-LP, including the modulation of alternative splicing [58], transcription factor recruitment [59], and cell survival [59].

The EBNA3 proteins are transcriptional regulators that include EBNA3A, EBNA3B, and EBNA3C [60]. EBNA3A and 3C are required for B-cell transformation [61] while 3B is not [62]. Interestingly, this family of proteins has conflicting roles in oncogenesis, with EBNA3A and 3C promoting carcinogenesis and 3B inhibiting it [63]. EBNA3A promotes cancer formation through mechanisms such as stimulating cell proliferation via repressing p21WAF1/CIP1, a cyclin-dependent kinase inhibitor [64], targeting tumor suppressor pathways [65], and altering cell cycle regulation [65]. EBNA3C promotes oncogenesis via multiple mechanisms including targeting tumor suppressor proteins [65] and disrupting cell cycle progression [66]. In addition, EBNA3A and 3C combine to prevent B cells from differentiating into plasma cells, which contributes to the establishment of long-term latent infection and lymphomagenesis [63]. EBNA3B is a tumor suppressor; indeed, mouse studies showed that B cells infected with an EBNA3B-deficient strain of EBV formed highly aggressive, DLBCL-like tumors. Cells infected with an EBV strain lacking EBNA3B demonstrated decreased T-cell-mediated killing secondary to the decreased secretion of the chemoattractant CXCL10. These tumors also proliferated more quickly [67].

The transformative potential of LMP1 was first recognized several decades ago when its expression was noted to stimulate anchorage-independent growth and to abrogate contact inhibition in RAT1 cells. This early experiment noted that LMP1-expressing cells were tumorigenic [68]. Since then, it has been recognized as a key oncogene that mimics CD40 signaling in B cells [34] and is critical for B-cell transformation [69]. Interestingly, one study showed that treatment of EBV^+^ cells with N-acetylcysteine amide (NACA) reduced LMP1 expression. Furthermore, cells treated with NACA had diminished cell division and lymphoblastoid cell line (LCL) outgrowth [70]. Furthermore, LMP1 has been associated with the increased expression of PD-L1 in multiple lymphoid malignancies. LMP1^+^ cells have demonstrated increased PD-L1 expression when compared to LMP1^−^ cells [71]. This has significant implications regarding therapeutic options for EBV-associated malignancies [72,73]. LMP1 has multiple other functions, including cytokine/chemokine induction, tumor angiogenesis, and immune modulation [69]. The variety of roles played by LMP1 can be explained by its extensive capacity to alter cell pathway regulation. For example, it has been shown to upregulate molecular signaling pathways such as NF-κB [74], Jak/STAT [75], EGFR [76], and ERK [76,77].

LMP2A is best known to mimic B-cell receptor (BCR) signaling. At the same time, it blocks BCR signaling to prevent lytic replication [78,79]. Mechanistically, it has been shown to mediate this effect through its immunoreceptor tyrosine activation motif (ITAM). Mutations to the ITAM allowed for tyrosine phosphorylation, calcium mobilization, and BZLF1 induction following BCR cross-linking. In addition, Syk protein tyrosine kinase (PTK) could no longer bind LMP2A in cells infected with the mutant virus strains [78]. Subsequent experiments confirmed LMP2A mimicry of a multitude of BCR signaling events, including phosphorylation of SYK, the BLNK, BTK, and PLCγ2 complex, and NFAT [79]. However, its impact extends beyond that of a BCR mimic. In fact, it was shown to regulate a slew of cellular processes that promote cellular survival and proliferation, including cell cycle progression, apoptosis, and proliferation [79].

LMP2B promotes the transition from latent to lytic replication. LMP2B overexpression resulted in the increased expression of lytic genes and reduced the amount of BCR stimulation required to transition to the lytic cycle in one study [80]. LMP2B has also been implicated in epithelial cell spread [81]. A more comprehensive discussion of EBV latent proteins can be found elsewhere [39,40,55,65,69,81].

In essence, EBV expresses latent proteins that play key roles in the pathogenesis of EBV infection. Furthermore, they play essential roles in carcinogenesis. These proteins are responsible for a diverse variety of functions ranging from cell transformation/cell reprogramming to immune evasion, immune suppression, angiogenesis, cell cycle regulation, and BCR mimicry.

## 3. EBV and Oncogenesis

EBV is associated with numerous subtypes of lymphoma [82,83]. The following section will discuss EBV’s contribution to lymphomagenesis.

### 3.1. Diffuse Large B-Cell Lymphoma

EBV^+^ diffuses large B-cell lymphoma (DLBCL); not otherwise specified (NOS) is an aggressive lymphoma variant with a relatively poor prognosis [84]. This entity was originally associated with the elderly [85], leading to the hypothesis that the immune senescence that naturally occurs with aging may play a role in its oncogenesis [86]. The aging immune system develops many changes, including but not limited to chronic low-level inflammation, a diminished ability to handle pathogens, and an increased risk of autoimmunity and cancer [87]. DLBCL has since been recognized as an entity that can affect younger individuals as well [88,89].

EBV^+^ DLBCL typically expresses a type III latency program [84], and EBNA2 has been shown to play a role in its oncogenesis. One way in which this is accomplished is through the induction of PD-L1. Notably in one study, when EBNA2 expression was induced by infecting the EBV negative DLBCL cell lines U2932 and SUDHL5 with EBV, PD-L1 expression was upregulated in both lines. When tested in EBNA2-transfected U2932 cells, miR-34a, a cellular miRNA, was downregulated [53]. Notably, miR-34a expression is inversely correlated with PD-L1 expression [90]. miR-34a overexpression in this infected cell line resulted in significant tumor cell death, indicating that the EBNA2-mediated downregulation of miR-34a and subsequent upregulation of PD-L1 suppresses the immunogenicity of infected cells. Consistent with these results, EBV^+^ DLBCL cells obtained from patients demonstrated higher PD-L1 expression. The increase in PD-L1 expression in EBNA2^+^ samples when compared to EBNA2^−^ samples was particularly conspicuous when assessing the number of cells that stained strongly for PD-L1 [53].

S1PR2 Is a sphingosine receptor that has been shown to prevent B-cell migration out of germinal centers [91]. Indeed, S1PR2 signaling is inactivated in DLBCL [92]. A recent set of experiments has implicated LMP1 in dysregulated S1PR2 signaling. Samples of EBV^+^ DLBCL with intact LMP1 expression were more likely to lack S1PR2 than tumors not expressing LMP1. Downstream effects of the reduced S1PR2 expression include constitutive PI3K path expression [93]. In mouse models, LMP1 has also been shown to cooperate with Ebf1 and Rel to promote lymphomagenesis. Following the induction of LMP1 expression in CD43^−^ B cells from R26LMP1^stopf^ mice, retroviruses carrying either Ebf1 or Rel genes were introduced into the cell. LMP1-expressing cells transformed into mouse lymphoblastoid cell lines with the addition of either Ebf1 or Rel, but not with the control. The authors further showed that loss of plasma cell differentiation secondary to Ebf1 activation, or as yet undiscovered events in Rel^high^ cells, is a key element of LMP1 lymphomagenesis [94]. Notably, LMP1 was previously implicated in preventing B-cell differentiation into plasma cells by downregulating BLIMP1α [95].

Murine experiments also demonstrated that LMP1 collaborates with LMP2A to promote early lymphomagenesis, but that neither protein is absolutely required. Working with Humanized NOD/LtSz-scid/IL2Rγ^null^ mice, Ma et al. demonstrated that deleting LMP1 and LMP2A simultaneously did not inhibit the development of lymphomas. However, the frequency with which lymphomas developed was reduced, and tumors took longer to develop [96].

Nagel et al. recently demonstrated that NKL homeobox gene activity is dysregulated in DLBCL [97]. Expression profiling of DOHH-2 cells, a DLBCL line with <5% EBV positivity, unsurprisingly demonstrated altered expression patterns of several B-cell genes between the EBV^+^ and EBV^−^ subsets, including the downregulation of BCL6, BACH2, and IL4R and increased levels of IRF4, MIR155HG, and PRDM1. The NKL homeobox genes HLX, NKX6-3, and MSX1 also demonstrated altered expression profiling in DOHH-2 cells, a difference that was particularly notable in HLX and NKX6-3 cells. Subsequent experiments demonstrated that EBV-mediated STAT3 upregulation increased HLX activity. Notably, it was demonstrated that HLX inhibits B-cell differentiation factors SPIB and IL4R. The authors further showed that EBV inhibited apoptosis in the presence of etoposide, potentially through the activation of HXL and subsequent downregulation of the pro-apoptotic gene BCL2L11/BIM [98].

EBV^+^ DLBCL expresses increased levels of NF-κB when compared to EBV^−^ DLBCL [99,100]. Indeed, nuclear staining for the NF-κB subunits p105/p50 and p100/p52 showed that 79% of cases were positive for the former, 74% for the latter, and 63% were positive for both subunits. These results are consistent with NF-κB activation through both the classical and alternative pathways. The authors concluded that EBV may play a role in activating the NF-κB pathway in DLBCL [99].

In summary, EBV latent proteins stimulate a wide variety of changes that promote the formation of DLBCL, such as the upregulation of PD-L1, which has major implications in treatment. Management will be discussed in the next section. It also stimulates alteration in S1PR2 signaling, which has significant downstream effects, and upregulation of STAT3. These changes cumulatively promote the formation of DLBCL.

#### Management

Treatment for DLBCL includes rituximab (anti-CD20 monoclonal antibody), cyclophosphamide, doxorubicin, vincristine, and prednisone (R-CHOP R-CHOP has demonstrated less significant responses in EBV^+^ DLBCL than in EBV^−^ DLBCL patients [7]. Currently, there are no accepted treatments targeting EBV^+^ DLBCL, and the development of such targeted treatment remains elusive [101,102]. A phase II trial used the knowledge that LMP1 has a similar function to that of the BTK-dependent B-cell receptor and used ibrutinib, a BTK inhibitor in combination with R-CHOP. While effective, it did not demonstrate a significant improvement when compared to R-CHOP. It was, however, associated with serious toxicity and treatment-related death in older patients [103]. POLARIX, a double-blind phase III trial, evaluated a modified R-CHOP regimen (pola-R-CHP) where polatuzumub vedotin (anti-79b monoclonal antibody) replaced vincristine. Patients receiving pola-R-CHP had lower rates of disease progression/relapse than those receiving R-CHOP [104]. Bortezomib is another agent that, when tested in mice, induced the apoptosis of EBV-transformed B cells and prolonged survival following the inoculation of those mice with the afore-mentioned transformed B cells [105]. Lenalidomide exhibited antitumor activity against DLBCL cells, particularly ABC-DLBCL cells. It was postulated that this difference arises through the inhibition of IRF4 expression and BCR-NF-κB signaling pathway in a cereblon-dependent manner [106]. PD-L1 is associated with a poor prognosis in aggressive lymphomas, and PD-L1 is commonly upregulated in EBV^+^ lymphoproliferative disorders (LPD). Nivolumab, a fully human IgG4 monoclonal anti-PD-1 antibody, is both safe and effective in lymphoid malignancies. A phase II study including nine patients hypothesized that PD-1 blockade may treat EBV^+^ LPD and NHL by reversing tumor-specific T-cell inactivation. ORR was 60% (three out of five) and CR was 40% (two out of five) in the LPD group [107]. The adoptive transfer of antigen-specific T lymphocytes is emerging as a potential avenue for the treatment of virus-associated malignancies, including EBV-associated malignancy [108]. In a cohort of 101 patients assessing anti-CD19 chimeric antigen receptor (CAR) T-cell therapy, there was an objective response rate of 76%. Overall, 47% of patients experienced a complete response, and progression-free survival at 1 month was 92%. It was 53% at 3 months [109]. A trial by Wudhikarn et al. assessed the efficacy of CAR T-cell therapy in relapsed/refractory DLBCL in patients who had no evidence of disease at the time of CAR T-cell treatment. A total of 24 patients received tisagenlecleucel and 9 received axicabtagene ciloleucel. Event-free survival was 59.6% and overall survival was 81.3% [110]. There are several ongoing studies for the treatment of EBV^+^ DLBCL, such as the NAVAL-1 (nanatinostat in combination with valganciclovir) phase II trial. Nanatinostat targets specific class I HDAC isoforms. Class I HDAC suppression results in the activation of the lytic gene BGLF4 and subsequent activation of ganciclovir [111,112]. The combination of nanatinostat and valganciclovir was further analyzed by Haverkos et al. in a phase Ib/II trial that showed a 67% overall response rate, including a 33% complete response rate in DLBCL [112]. Another study seeking to develop EBV-directed therapy is a phase II, prospective, multi-center study assessing the combination of Sintilimab (an anti-PD-1 antibody) and R-CHOP in previously untreated patients with EBV^+^ DLBCL, NOS [113].

### 3.2. Classic Hodgkin Lymphoma

Hodgkin lymphoma is a malignancy characterized by the presence of Hodgkin cells, which are mononucleated, and Reed–Sternberg cells, which are binucleated. These cells express clusters of differentiation (CD) 20, 30, and occasionally CD15. They are thought to originate from pre-apoptotic germinal center B cells [114]. Notably, these cells are relatively rare compared to the inflammatory tumor microenvironment in which they exist [115]. EBV’s relation to such cells has long been known [116]. Indeed, it is estimated that approximately 30% of classical Hodgkin lymphoma (cHL) cases in the Western world are associated with EBV [117], and recent research has shown that the lymphomagenesis of a subset of EBV^−^ lymphomas may in fact be related to EBV [118]. Notably, this statistic does not reflect the nodular lymphocyte-predominant variant of Hodgkin lymphoma, which can have EBV^+^ lymphocyte-predominant cells, but is not typically EBV-associated [119].

As was previously mentioned, Hodgkin lymphoma is characterized by a type II latency program [39]. LMP1 plays a key role in lymphomagenesis, in part through the reprogramming of germinal center (GC) B cells so they more closely resemble a Hodgkin Reed–Sternberg cell (HRS) phenotype [120]. Of 20,551 GC B-cell genes examined on the HG-U133 plus microarray, LMP1 upregulated 622 and downregulated 1304. Common pathways with altered expression in both LMP1-infected cells and HRS cells include NF-κB, AP-1/ATF, STAT, and PI3K as well as FAS/CD95 and anti-apoptotic genes. Pro-apoptotic genes were downregulated. A total of 881 genes were changed by both LMP1 and HRS cells [120]. LMP1 downregulated B-cell-specific genes and upregulated the B-cell-specific gene suppressor ID2 [120]. Loss of B-cell-specific genes is a hallmark of HRS cells [121]. Notably, many of the pathways upregulated by LMP1, such as NF-κB and JAK/STAT, are also upregulated in HRS cells [115]. LMP1 has additionally been shown to induce CD137 in HRS cells through the activation of the PI3K-AKT-mTOR signaling axis [122]. CD137 activity was previously shown to contribute to immune evasion in Hodgkin lymphoma by inhibiting T-cell activation through downregulation of CD137 ligand [123]. LMP1 expression was further demonstrated to correlate with the expression of the chemokines CCL17 and CCL22 production in HRS [124]. These chemokines have previously been shown to enhance regulatory T-cell (Treg) recruitment in various EBV-related malignancies [125,126]. In addition, LMP1 expression plays a role in regulating autophagy. Hodgkin lymphoma cells expressing LMP1 were better able to adapt to starvation-induced autophagic stress [127].

Alterations in the gene expression are a hallmark of lymphoma and of cancer in general [128,129,130]. LMP2A also induces changes in B-cell gene transcription that mimic Reed–Sternberg cells [131]. Indeed, LMP2A was noted to result in the differential expression of 159 genes in LMP2A-expressing BJAB cells and 139 genes in EBV-infected LCLs. These changes impacted genes involved in multiple cellular processes including ubiquitination, transcription, and cell cycle/apoptosis. Interestingly, genes implicated in RNA/DNA were impacted in human cell lines, implying that LMP2A can cause deregulated DNA replication/gene transcription [131]. Furthermore, Anderson and Longnecker used a mouse model to demonstrate that LMP2A acts through the Notch1 pathway to impact B-cell development [132].

Another way in which LMP2A contributes to the pathogenesis of Hodgkin’s lymphoma is through its impact on the tumor microenvironment (TME) [133]. The TME in cHL is highly complex and plays many roles, including immune suppression, survival, and proliferation. One mechanism by which this occurs is by attracting immune cells into the TME [117]. LMP2A expression was highly correlated with the expression of the cytokine MIP-1α [133]. Cells containing mutant EBV whose LMP2A did not bind SYK did not upregulate MIP-1α expression. The same set of experiments demonstrated that LMP2A upregulates MIP-1α expression through the activation of Syk/PI3K/NF-κB signaling [133].

EBNA1 has also been implicated in the pathogenesis of cHL, for example, through promoting the survival of Reed–Sternberg cells [134]. For example, protein tyrosine phosphatase receptor kappa (PTPRK) expression was shown to be downregulated in EBV infection. Mechanistically, it was determined that this downregulation was accomplished through the downregulation of SMAD2. These results are significant because PTPRK is a tumor suppressor protein. Indeed, the downregulation of PTPRK mRNA and protein resulted in a significant increase in both cell viability and proliferation. EBV^+^ cHL tumor cells demonstrated more frequent downregulation of PTPRK than EBV^−^ cHL tumor cells [135]. Furthermore, EBV increases the expression of CCL20 to promote Treg migration. This allows EBV to recruit Tregs to the TME, which provides a mechanism by which EBV-infected cells can downregulate the anti-tumor immune response [136].

In summary, EBV promotes a rather dramatic change in B cells to promote the pathogenesis of cHL, including a wholesale reprogramming of genetic expression to more closely resemble a HRS phenotype. Changes include alterations in numerous pathways such as anti-apoptotic paths, FAS/CD95, and STAT. It also plays a role in immune evasion. LMP2A contributes to the remodeling of B cells to more closely mimic the transcription pattern of HRS cells. It also alters the TME, which has numerous critical roles in immune suppression, survival, and proliferation. EBNA1 also contributes to cHL pathogenesis, for example, through its pro-survival effects on HRS cells.

#### Management

The initial treatment of Hodgkin lymphoma depends on disease stage and prognostic factors. Patients are divided into three groups: patients with early-stage disease with favorable prognostic factors, patients with limited-stage disease and unfavorable prognostic factors, and those with advanced-stage disease [137]. Combination chemoradiotherapy remains integral to the treatment of patients with early disease and either favorable or unfavorable factors. ABVD (doxorubicin, bleomycin, vinblastine, dacarbazine), BEACOPP (bleomycin, etoposide, doxorubicin, cyclophosphamide, vincristine, procarbazine, prednisone), and MOPP (nitrogen mustard, vincristine, procarbazine, prednisone) are some of the chemotherapy regimens used along with radiation depending on disease stage. Patients with advanced disease receive chemotherapy alone [137]. Detecting EBV within cHL may help confirm diagnosis but does not influence therapeutic selection. [18,115]. In general, outcomes are poor in the elderly [138], and an EBV^+^ status is associated with even worse prognosis [139]. EBV infection constitutes another means for PD-L1 induction in cHL, and PD-1 blockade represents a possible avenue for targeted therapy. [140]. Brentuximab vedotin, an anti-CD30 antibody, is another therapy that has been examined in the setting of cHL [141]. For example, a recent phase I–II trial examined this drug in combination with nivolumab in the setting of relapsed/refractory cHL. Patients could receive an autologous stem cell transplant (ASCT), but did not have to, following therapy. The objective response rate was 85%, including a 67% complete response rate. Overall survival was 93% at 3 years [142]. The phase II HOVON/LLPC Transplant bRaVE trial assessed its efficacy in relapsed/refractory cHL in combination with dexamethasone, high-dose cytarabine, and cisplatin. Patients with partial/complete response went on to receive an autologous peripheral blood stem-cell transplant. 42 of 52 patients experienced a metabolic complete response prior to transplant while another 5 had a partial response prior to transplant. Progression free survival was 74%, and overall survival was 95% at 2 years [143]. Brentuximab vedotin’s role in stage III/IV cHL has also been examined. The Echelon-1 study demonstrated that brentuximab vedotin combined with doxorubicin, vinblastine, and dacarbazine demonstrated greater clinical efficacy than ABVD, with a 3-year progression-free survival of 83.1% vs. 76.0%, respectively [144]. CAR T cells have likewise been shown to have potential as a therapeutic modality in cHL [145]. For example, Ramos et al. assessed its efficacy in patients with previously treated cHL. Patients had received an average of 7 previous lines of chemotherapy. Of the 32 patients analyzed, overall response was 72%, including a 59% complete response rate. 1 year progression-free survival was 36%, with an overall survival rate of 94% [146]. Another trial assessed the results of autologous stem cell transplantation in combination with anti-CD30 CAR T-cell therapy. 4 of 5 cHL patients experienced a complete response with the other experiencing a partial response. Responses were durable through the median follow up of 20.4 months [147]. Other therapies under study include vaccine approaches that have been developed based on recombinant virus proteins or virus peptides [148], EBV-specific cytotoxic T lymphocytes (CTLs) [148], therapies targeting specific EBV latent proteins [149], and EBNA1 inhibitors [150].

### 3.3. Burkitt Lymphoma

Burkitt lymphoma is a non-Hodgkin lymphoma that predominantly affects children. It is most strongly associated with a MYC oncogene translocation, although multiple other mutations play key roles as well, including those affecting TP53, ARF, and DDX3X [151]. The association between Burkitt lymphoma and EBV has long been known; indeed, it is the first cancer found to have viral contributions to its pathogenesis [151,152]. There are three forms: endemic, sporadic, and AIDS-Burkitt lymphoma. All three types can be associated with EBV [153]. Notably, at least for the endemic form, malaria is also involved in the development of Burkitt lymphoma [154].

Burkitt lymphoma is primarily associated with a type I latency pattern, meaning EBNA1 is the main protein expressed [37]. Indeed, there is some evidence that EBNA1 may be involved in the pathogenesis of Burkitt lymphoma [155]. For example, EBNA1 knockout using the transcription activator-like effector nuclease E1TN resulted in the death of cells infected by EBV. Furthermore, it was noted to cause progressive loss of EBV episomes from Burkitt lymphoma cells infected by EBV [156]. Wang et al. recently examined the role of EBNA1 in the pathogenesis of Burkitt lymphoma [157]. Vav1, a proto-oncogene, is aberrantly expressed in a multitude of B-cell malignancies, including non-Hodgkin lymphoma [158]. This molecule impacted apoptosis in the EBV^+^ cell lines RAJI and LCL-1 but not the EBV^−^ cell line BJAB [157]. Cell lines with decreased Vav1 expression had increased levels of the pro-apoptotic BCL-2 protein Bim. It was subsequently shown that Vav1 binds to EBNA1 and that EBNA1 expression was inversely correlated with Bim expression. The implication is that the interaction between EBNA1 and Vav1 results in enhanced resistance to apoptosis through suppression of Bim [157]. EBNA3A and EBNA3C have also been shown to play a role in Bim suppression. This suppression was noted even in the setting of latently infected cells. EBNA3A and EBNA3C appear to accomplish this through downregulating Bim RNA. The survival advantage conferred by EBNA3A and EBNA3C was negligible in the absence of Bim [159].

EBV-associated Burkitt lymphoma has differential gene expression when compared to EBV^−^ Burkitt lymphoma. It was recently shown that EBV miRNAs impact the molecular profile of Burkitt lymphoma. Indeed, 103 genes known to be viral miRNA targets were differentially expressed in EBV^+^ tumors when compared to EBV^−^ tumors. These genes were involved in many key cellular processes, including transcription, gene expression, nucleotide/RNA metabolism, and apoptosis. Several were related to tumorigenesis, including TP53, TGFB, mTOR, and others besides [160]. The substantial impact of miRNAs corroborates studies demonstrating key roles for these molecules in Burkitt lymphoma cell survival [161]. Ectopic BART miRNA introduced into Burkitt lymphoma cell lines resulted in decreased apoptosis and promoted cell proliferation. It was shown that BART miRNAs target transcripts of CASP3 [161], a gene involved in apoptosis [162]. Subsequent experiments demonstrated that the 12 BARTs inhibited CASP3 transcripts, with BART22 responsible for the most significant repression [163]. Moreover, BART miRNAs stimulated B-cell transformation as well as survival and/or proliferation [161]. Furthermore, BARTs have been shown to inhibit host defenses in the setting of Burkitt lymphoma. For example, overexpression of miR-BART-6-3P resulted in a decreased expression of IL-6R. Furthermore, it acted synergistically with the cellular microRNA miR-197, which also targets IL-6R, to further downregulate IL-6R [164]. miR-BART-6-3P also acts synergistically with miR-142 to reduce the expression of both IL-6R and PTEN [165], a tumor suppressor that inhibits the PI3K/AKT/mTOR pathway [166].

In summary, Burkitt lymphoma is associated with type I latency. As such, EBNA1 is key to its pathogenesis. EBNA1 knockout results in the apoptosis of EBV-infected cells and leads to episomal loss in Burkitt lymphoma cells. Furthermore, it results in aberrant VAV1 expression, which ultimately enhances cell survival. EBV miRNAs significantly alter the gene expression of multiple cell processes including apoptosis, transcription, and gene regulation.

#### Management

The endemic subtype of Burkitt lymphoma is known to be universally associated with EBV, and EBV is detected in 25% to 40% of sporadic and immunodeficiency-associated cases [167]. Burkitt lymphoma is highly chemotherapy sensitive [167], and the commonly used regimens in the US include rituximab, etoposide, prednisone, vincristine, cyclophosphamide, and doxorubicin (R-EPOCH); cyclophosphamide, doxorubicin, vincristine, methotrexate, ifosfamide, cytarabine, and etoposide (CODOX-M/IVAC); hyperfractionated cyclophosphamide, vincristine, doxorubicin, dexamethasone alternating with high-dose methotrexate, and cytarabine (hyper-CVAD) [167,168]. Low-dose adjusted R-EPOCH facilitates the treatment of elderly and HIV-positive patients by reducing the toxicity associated with higher intensity therapy. [169]. In patients with advanced disease, autologous and allogeneic hematopoietic stem cell transplant are used as salvage therapy followed by second-line chemotherapy for disease control [170]. Patients refractory to primary treatment have a poor prognosis [171]. Autologous and allogeneic hematopoietic stem cell transplant are used as salvage therapy with limited efficacy [170]. Abraham Avigdor et al. reported a single case of Burkitt lymphoma refractory to multiple chemotherapy regimens and allogenic HCT, where a good response was achieved with salvage CAR T-cell therapy. Given the response, this treatment option could be considered for those with relapsed/refractory disease. [172]. Dalton et al. reported the use of low-dose decitabine, a hypomethylating agent. as a novel epigenetic therapeutic agent that can sensitize weakly immunogenic (i.e., EBV^+^) Burkitt lymphoma expressing type I latency by inducing the expression of more immunogenic proteins such as LMP1, EBNA2, EBNA3A, and EBNA3C. Induction of these proteins allows for EBV-specific T cells to locate and lyse tumor cells [173].

### 3.4. NK/T-Cell Lymphoma

NK/T-cell lymphoma is a type of non-Hodgkin lymphoma most frequently seen in Latin America and Asia [174]. The predominant type is extranodal NK/T-cell lymphoma, nasal type, although other pathologies exist [175]. Mutations are common, particularly TP53, DDX3X, STAT3, JAK3, MGA, BCOR, ECSIT, and MCL1 [174]. These tumors are strongly associated with EBV [174,175], although its precise role in generating these tumors is not well explored [176].

One way in which EBV contributes to the pathogenesis of NK/T-cell lymphoma is through the actions of LMP1. LMP1 activates PGC1β, a member of the peroxisome proliferator-activated receptor-γ (PPARγ) coactivator-1 (PGC1) family, via activation of NF-κB [177]. PGC1β has previously been implicated in tumorigenesis [178,179]. mRNA and protein levels of PGC1β and its downstream target 8-oxoguanine DNA glycosylase (OGG1), a base excision repair enzyme, were significantly increased in multiple NK/T-cell lines. Mechanistically, it was shown that this effect was mediated by NF-κB activation, which subsequently regulated the PGC1β promoter [177]. Furthermore, the disruption of the signaling pathway through the interruption of hexokinase domain component 1 (HKDC1) caused mitochondrial dysfunction and led to the generation of ROS. This led to EBV suppression [177,180]. The role of LMP1 was further explored by Sun et al. [181], who showed that short hairpin RNA (shRNA) targeting LMP1 reduced host migration and invasion. It also promoted apoptosis in the human EBV^+^ NKTL cell line SNK-6. The authors showed that the LMP1-mediated activation of NF-κB results in the increased expression of elF4E [181], a molecule that has been associated with a poor prognosis in multiple malignancies [182,183]. The LMP1/NF-κB/elF4E axis then increases proliferation, migration, and invasion [181]. LMP1 has also been shown to decrease the expression of miR-15a. Suppression of miR-15a allows for increased expression of MYB and cyclin D1 and subsequent increased cell proliferation [184].

Interestingly, LMP1 expression in NK/T-cell lymphomas appears to be linked to the TME; monocytes co-cultured with nasal NK/T-cell lymphoma cells stimulated both LMP1 expression and cell proliferation through contact-dependent interactions. This effect was mediated by IL-15 as was evidenced by the fact that treatment with IL-15 antibodies abrogated this effect [185].

In summary, LMP1 upregulates PCG1β, which results in the upregulation in the base excision repair enzyme OGG1. Alterations in this signaling pathway led to mitochondrial dysfunction and subsequent ROS activation. In addition, LMP1 plays roles in migration, invasion, cell survival, and cell cycle progression. It also plays a role in manipulating the TME.

#### Management

Traditional therapies targeting B-cell lymphomas are largely ineffective in NK/T-cell lymphomas [186]. The majority of effective therapies include asparaginase [186], a compound shown to cause apoptosis in NK/T cells [187], or its pegylated form [186]. Treatment typically involves chemoradiotherapy [186,188]. There are multiple potential chemotherapy regiments recommended by the NCCN guidelines, including SMILE (steroid (dexamethasone), methotrexate, ifosfamide, pegaspargase, and etoposide), P-GEMOX (gemcitabine, pegaspargase, and oxaliplatin), and DDGP (dexamethasone, cisplatin, gemcitabine, and pegaspargase) [188]. New research has examined EBV as a potential therapeutic target [189]. In a phase II trial, 15 patients were treated with baltaleucel-T, a preparation of autologous cytotoxic T lymphocytes specific for EBV. Of the 15 patients, 10 had apparent disease prior to treatment initiation (salvage cohort), and the other 5 had no apparent disease (adjuvant cohort). Of the 10 patients in the salvage cohort, 2 had a partial response and 3 had a complete response. Progression-free survival in this group was 12.3 months on average. two of the five patients in the adjuvant cohort did not experience disease relapse. Additionally, there was no apparent difference between the responders and non-responders in plasma EBV DNA level [189].

Additional studies have examined T-cell therapy targeted towards LMP. While patients with T-cell-mediated disease (2 year overall survival (OS) 60%) did not fare as well as patients with B-cell-mediated disease (2 year OS 80%) [190], the data did compare favorably to the Center for International Blood and Marrow Transplant Research (CIBMTR) data (2 year OS 36%). In this latter study, patients received allogeneic hematopoietic stem cell transplant therapy for extranodal NK/T-cell lymphoma [191]. While the data were not statistically significant, patients responding to therapy tended to have received T cells with a greater specificity for LMP2/EBV antigens and for those who had higher levels of LMP2-specific T cells [190]. A separate trial examined the effectiveness of LMP1 and LMP2A-specific cytotoxic T lymphocytes in a cohort of 10 patients who demonstrated a complete response to therapy. OS at 4 years was 100%, and progression-free survival at 4 years was 90%. Furthermore, levels of IFN-γ-secreting T cells specific for LMP1 and LMP2A were inversely proportional to plasma levels of EBV DNA [192].

### 3.5. Primary CNS Lymphoma

Primary CNS lymphoma (PCNSL) is a fairly rare and lightly studied tumor limited to the brain, eyes, and cerebrospinal fluid. The vast majority of PCNSLs are DLBCL [193], although other histologies have been reported [194]. EBV plays a role in the development of a subset of these cases [12]. EBV-associated PCNSL was recently shown to differ from PCNSL not associated with EBV infection. EBV-associated PCNSL was significantly less likely to have mutations. Mutations commonly seen in EBV negative PCNSL, such as Myd88, PIM1, and CD79B, were not seen in EBV-associated PCNSL. EBV-associated tumors were also much less likely to be ABC-DLBCL [12]. This finding was confirmed by Radke et al. [195], who found HLA-DRB and immunoglobulin loci to be the only frequent mutations seen in EBV^+^ cases. Notably, EBV status has a marked impact on survival, with EBV^+^ patients having a significantly higher risk of death [196]. In summary, while more research is required, EBV has been linked to primary CNS lymphoma. Evidence supporting this fact includes differential mutation types and rates between EBV^+^ and EBV^−^ CNS lymphomas. EBV status also has significant implications in overall survival. Figure 2 provides a summary of how EBV latent proteins impact lymphomagenesis.

#### Management

EBV-associated PCNSL in the immunosuppressed is its own separate entity compared to EBV^−^, HIV^−^ PCNSL. Prospective therapies ideally are capable of penetrating the blood–brain barrier, interrupting EBV-mediated oncogenesis, and targeting EBV antigens [12]. The regimens used in non-CNS lymphoma, such as R-CHOP, are ineffective in PCNSL due to inadequate penetration of the blood–brain barrier [197]. Current guidelines suggest that rituximab and high-dose methotrexate should be a part of induction therapy [193]. The different regimes used for induction include R-MVP (rituximab, HD-MTX, vincristine, and procarbazine; R-MT (rituximab, HD-MTX, and temozolomide); MATRix (thiotepa, rituximab and HD-MTX, high-dose cytarabine) [193]. Relapsed/refractory cases have poor prognosis, and no optimal therapy has been established [193,198]. Currently, MTX salvage therapy is used, but several prospective studies involving combination regimens (rituximab and temozolomide) [199] and single agents such as pemetrexed, topotecan, temozolomide, and rituximab have shown acceptable objective response rates [200,201,202,203]. Ibrutinib (BTK-inhibitor) has shown impressive phase I efficacy in recurrent/refractory PCNSL [198]. Using EBV^+^ blood donors, Haque et al. generated EBV-specific cytotoxic T lymphocytes to treat EBV^+^ PTLD with promising results [204]. It is likely that EBV-specific CTL can be utilized in treating other EBV-associated malignancies [205]. A phase I Australasian Leukemia/Lymphoma Group clinical trial is also examining the possibility of incorporating EBV-specific third-party T cells (ACTRN12618001541291) in treatment regimens [12]. Slobod et al. had promising results treating two patients with HIV-associated PCNSL [206]. Table 1 includes a summary of therapeutic options for the EBV-related malignancies discussed above.

## 4. Conclusions

The clinical importance of EBV cannot be underestimated. It is associated with a multitude of malignancies of both hematologic and epithelial origin. Lymphoid neoplasms are prominently featured among the hematologic malignancies, including DLBCL, cHL, Burkitt lymphoma, primary CNS lymphoma, and NK/T-cell lymphoma.

Treatment for these EBV-associated diseases is progressing. However, EBV is still implicated in hundreds of thousands of new cancer cases per year and nearly 2% of all cancer deaths. An improved understanding of the virus will allow for novel therapies that could both enhance treatment and prevent the development of a multitude of diseases. Indeed, advances are already underway. For example, both prophylactic and therapeutic vaccines targeting EBV are under examination [207], and compounds are being assessed for their antiviral efficacy [208]. However, more research is required to better address the global health challenges posed by this virus.

## Figures and Tables

**Figure 1 cancers-15-02133-f001:**
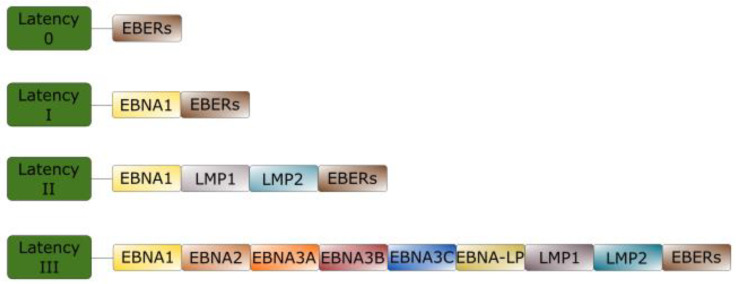
Patterns of gene expression in Epstein-Barr virus (EBV) latency. Following initial infection, EBV establishes 4 common patterns of gene expression termed type 0 latency, type I latency, type II latency, and type III latency. No proteins are expressed in type 0 latency. In type I latency, Epstein–Barr nuclear antigen (EBNA) 1 is the only protein expressed. In type II latency, EBNA1, latent membrane protein (LMP) 1, and LMP2 are expressed. Type III latency features the expression of all EBV-associated latency proteins including EBNA1, EBNA2, EBNA3, EBNA-LP, LMP1, and LMP2. Notably, Epstein–Barr virus-encoded small RNAs (EBERs) are expressed in all forms of latency.

**Figure 2 cancers-15-02133-f002:**
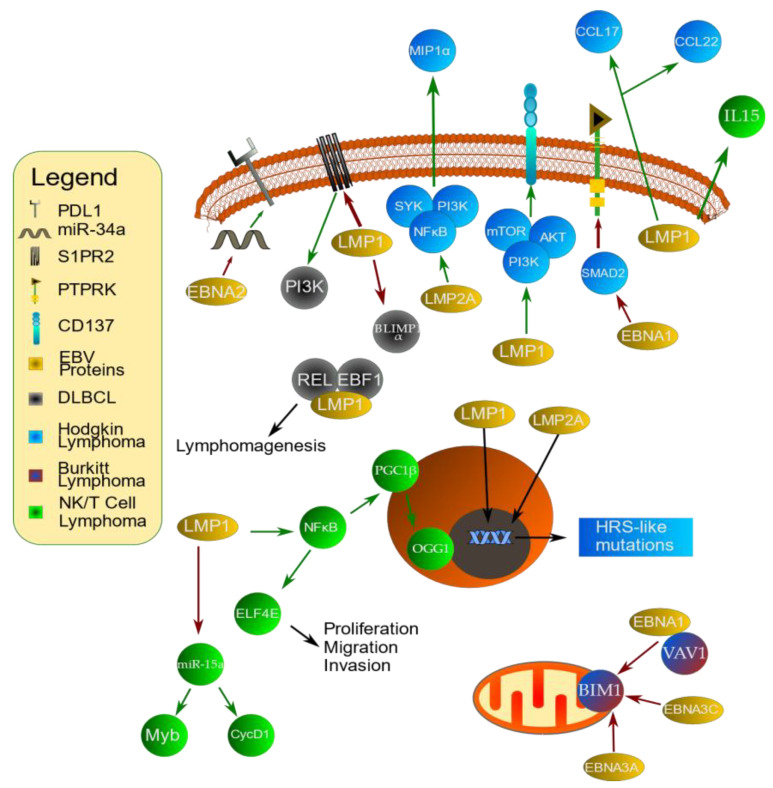
EBV-associated latent proteins and tumorigenesis. EBV latent proteins are strongly associated with tumorigenesis. For example, in diffuse large B-cell lymphoma (DLBCL), represented by dark gray in the figure, LMP1 inhibits BLIMP1 α and S1PR2. S1PR2 inhibition allows for increased PI3K signaling. It also cooperates with REL and EBLF1 to promote lymphomagenesis. EBNA2 inhibits miR-34a, which allows for increased expression of PDL1. In Hodgkin lymphoma, represented by blue in the figure, LMP1 stimulates CD137 activity through the PI3K/AKT/mTOR pathway. It also promotes expression of the chemokines CCL17 and CCL22. Furthermore, it promotes mutations that cause cells to more closely resemble HRS cells. LMP2A stimulates increased signaling through Syk/PI3K/NF-κB, which results in increased expression of the cytokine MIP-1α. EBNA1 inhibits SMAD2, which in turn decreases expression of the tumor repressor PTPRK. Activation of LMP1 in NK/T lymphoma, represented by green in the figure, increases expression of IL-15. It also mediates NF-κB activation, which in turn leads to increased expression of PGC1β and its target, the base excision repair enzyme OGG1. NF-κB also increases ELF4E expression, which promotes proliferation, migration, and invasion. Lastly, it inhibits miR-15a, which leads to increased Myb and cyclin D1 expression. Green arrows in the figure indicate that the next step in the pathway is upregulated while red arrows indicate inhibition of the next step.

**Table 1 cancers-15-02133-t001:** Treatment for EBV-related lymphomas.

Cancer	Therapy	Notes	Citation
DLBCL	Rituximab, cyclophosphamide, doxorubicin, vincristine, prednisone (R-CHOP)	Rituximab: anti-CD20 monoclonal antibody Standard of care	[7]
	R-CHOP + ibrutinib	Ibrutinib: BTK inhibitor included because LMP1 mimics BTK-dependent B-cell receptor	[103]
	Rituximab, cyclophosphamide, doxorubicin, polatuzumab vedotin, prednisone (pola-R-CHP)	Polatuzumab vedotin: anti-79b monoclonal antibody	[104]
	Bortezomib	Apoptosis of EBV-transformed B cells in mice	[105]
	Lenalidomide	Inhibits IRF4 and BCR-NF-κB	[106]
	Nivolumab	IgG4 monoclonal antibody targeting PD-1	[107]
	Antigen-specific T cells		[108,109,110]
	Nanatinostat + valganciclovir	Nanatinostat: selective for isoforms of class I HDACs	[111,112]
	R-CHOP + Sintilimab	Sintilimab: anti-PD-1 antibody	[113]
Classic Hodgkin Lymphoma	Doxorubicin, bleomycin, vinblastine, dacarbazine (ABVD)	Standard of care, can be used in combination with radiation	[137]
	Bleomycin, etoposide, doxorubicin, cyclophosphamide, vincristine, procarbazine, prednisone (BEACOPP)	Standard of care, can be used in combination with radiation	[137]
	Nitrogen mustard, vincristine, procarbazine, prednisone (MOPP)	Standard of care, can be used in combination with radiation	[137]
	Anti-PD-1 pathway immune checkpoint inhibitors		[140]
	Brentuximab vedotin	Anti-CD30 antibody	[141,142,143,144]
	T-cell therapy		[145,146,147,148]
	EBV vaccination		[148]
	Therapies targeting EBV latent proteins/EBNA1 inhibitors		[149,150]
Burkitt Lymphoma	Rituximab, etoposide, prednisone, vincristine, cyclophosphamide, doxorubicin (R-EPOCH)	Standard of care Low dose adjusted form for elderly/HIV+ patients	[167,168,169]
	Cyclophosphomide, doxorubicin, vincristine, methotrexate, ifosfamide cytarabine, etoposide (CODOX-M/IVAC)	Standard of care	[167,168,169]
	Hyperfractionated cyclophosphamide, vincristine, doxorubicin, dexamethasone alternating with high-dose methotrexate and cytarabine (hyperCVAD)	Standard of care	[167,168,169]
	Autologous/allogenic hematopoietic stem cell transplant	Salvage therapy	[170]
	T-cell therapy	Case report of salvage therapy	[172]
	Decitabine	Epigenetic induction of immunogenic EBV proteins	[173]
NK/T-cell lymphoma	Asparaginase	Key component of therapy; causes NK/T cell apoptosis	[186,187]
	Steroid (dexamethasone), methotrexate, ifosfamide, pegaspargase, etoposide (SMILE)	Standard of care	[188]
	Gemcitabine, pegaspargase, oxaliplatin (P-GEMOX)	Standard of care	[188]
	Dexamethasone, cisplatin, gemcitabine, pegaspargase (DDGP)	Standard of care	[188]
	Balataleucel-T	Autologous EBV-specific CTLs; studied in salvage/adjuvant setting	[189]
	T-cell therapy targeting LMP		[190,192]
	Allogeneic hematopoietic cell transplantation		[191]
Primary CNS lymphoma	Rituximab + high-dose methotrexate	Included in induction therapy	[193]
	Rituximab, HD MTX, vincristine, procarbazine (R-MVP)	Induction chemotherapy	[193]
	Rituximab, HD MTX, temozolomide (R-MT)	Induction chemotherapy	[193]
	Thiotepa, rituximab, HD MTX, high-dose cytarabine (MATRix)	Induction chemotherapy	[193]
	Rituximab, temozolomide	Salvage chemotherapy	[199]
	Single agent salvage therapy: temozolomide, rituximab, pemetrexed, topotecan		[200,201,202,203]
	Ibrutinib	BTK inhibitor	[198]
	EBV-specific CTLs	Examined in PTLD, but may be applicable in CNS lymphoma	[12,204,205]
	Hydroxyurea	Case report	[206]

## Data Availability

All papers cited in this manuscript can be found on PubMed or Google Scholar.

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
