# Peer review of "EBV and Lymphomagenesis"

_cancers, 2023, doi:10.3390/cancers15072133_

Round 1

Reviewer 1 Report

This is a useful review on EBV, the pathogenesis it causes and the treatment options for EBV-associated hematological malignancies. There are a few issues that the authors could address.

Under management of cHL, the use of anti-CD30 antibodies should be discussed. It may also be interesting to refer to new experimental treatment approaches such as chimeric antigen receptors. Similary, under management of DLBCL, the use of CD19 CAR should be mentioned.

Line 146: “EBNA3C” instead of “EBBNA3C”

Line 148: Spell out which “two proteins:” are referred to

Line 159: The “+” in “EBV+” should be superscript. Similar for all other such terms.

Line 181: The two blocs of references should be combined.

Line 223: “Mouse experiments” or “Murine experiments” instead of “Mice experiments”.

Line 260: “B cells” instead of “B Cells”.

Line 283/284: reformulate “ .. thought to be related to originate ..”

Line 289: The “-” in “EBV-” should be superscript. Also, EBV- cases.

Line 3b1: “BJAB” instead of “Bjab”

Line 391: “EBV-negative” instead of “EBV negative”.

Table 1, first line: “anti-CD20” instead of “andti-CD20”.

 Table 1: left align text in table.

 Acknowledgments: if nobody is acknowledged, this section can be omitted.

Sometimes one and sometimes two blanc spaces are used between sentences. This should be standardized, ideally to one blanc space.

Author Response

Thank you for your comments; we really do appreciate them. We have made the following changes to the manuscript per your recommendations:

  1. Included a discussion of anti-CD30 antibodies in cHL management in lines 411-426.
  2. Included a discussion of CAR therapy in lines 426-434.
  3. Inlcluded a discussion of CD19 CAR for DLBCL in lines 298-307.
  4. Line 146 (now line 150): The extra 'B' in EBNA was deleted.
  5. Line 148 (now line 152): 'two proteins' was replaced with 'EBNA3A and 3C'. 
  6. Line 159 and other occurences: + and - were made superscripts where appropriate throughoug the manuscript.
  7. Line 181 (now line 191): the reference blocs were combined.
  8. Line 223 (now line 238): 'mice' was replaced with 'murine'.
  9. Line 260 (now line 283): the C in cells was made lowercase.
  10. Lines 283-284 (now line 323-324): 'be related to' was deleted. It now reads, "They are thought to originate from pre-apoptotic germinal center B cells [114]."
  11. Line 289 and similar instances: all + and - in the manuscript were made superscripts where appropriate. 
  12. Line 381 (now line 460): BJAB was capitalized.
  13. Line 391 (now line 470): EBV-negative was changed to EBV-.
  14. Table 1, line 1: 'andti' was edited to 'anti'.
  15. Table 1 was left text aligned.
  16. An acknowledgement was added.
  17. Instances of 2 spaces between words/sentences were changed to a single space.

Reviewer 2 Report

This review provides a comprehensive overview of the role of EBV latent proteins in the development of hematologic malignancies, including DLBCL, cHL, Burkitt lymphoma, primary CNS lymphoma, and NK/T cell lymphoma. The authors also provide a discussion of the management strategies for EBV-associated malignancies. 

It has been reported that LMP1 can also promote the expression of PD-L1. This review describes the role of PD-L1 in treating hematological malignancies and the relationship between EBNA2 and PD-L1 but does not mention the relationship between LMP1 and PD-L1.  Maybe the authors can add relevant content to the article. 

This review is mainly a combination of literature. Could you please add some summaries and your opinions?

There are several spelling problems in the text.

Line 146 “EBBNA3C”

Line 406 ”miR miR-197”

Line 412 ”The Endemic”

Line 486 You didn't put a point at the end of your sentence.

Overall, this review is a valuable resource for researchers and clinicians interested in understanding the role of EBV latent proteins in hematologic malignancies.

Author Response

Thank you very much for your comments; we appreciate them. We have made the following changes to improve our paper per your recommendations:

  1. Included the LMP1/PD-L1 relationshiop in lines 166-170.
  2.  Added non-citation material at the end of several sections (lines 192-196, 262-266, 383-390, 488-493. 560-564, and 608-611)
  3. Line 146 (now line 150): The extra 'B' in EBNA was deleted
  4. Line 406 (now line 484): spelled out microRNA. It now reads, "Furthermore, it acted synergistically with the cellular microRNA miR-197, which also targets IL-6R, to further downregulate IL-6R [164]." 
  5. Line 412 (now line 496): 'Endemic' is now 'endemic'
  6. Line 486 (now 587): A period was added at the end of the sentence.

Reviewer 3 Report

This manuscript provides a fairly comprehensive, concise review of the literature concerning the contributions of EBV to a variety of hematologic malignancies along with the latest therapies being employed for treating patients with these EBV-associated cancers.  It includes lots of the relevant literature published within the past 3 years and, thus, serves as a timely update to prior reviews in this field.

Minor suggestions:

1.       The text needs to be carefully read over and edited to correct multiple minor errors in wording, spelling, and grammar, e.g., on line 42, “medicated” should be “mediated”; on line 43, “no” such be “not”; on line 146, “EBBNA3C” should be “EBNA3C”; on lines 276-278, the statement is a phrase, not a complete sentence; line 284 needs to be reword; on line 295, “HRS” needs to define; reference 152 seems wrong for the statement made on lines 441-442;

2.       It may be better to shrink Figure 2 so that all of its legend can fit on the same page immediately below it.

3.       Table 1 – It would be nice to reformat this table so that it is more easily read while fitting on fewer pages.  This could be accomplished by making column 1 and, especially, column 4 smaller and columns 2 and 3 larger along with decreasing the spacing between the lines of text within boxes. 

Author Response

Thank you very much for your comments. The following changes were made to address them:

  1. Extensive spelling/grammar check was completed, including the suggestions you made in point 1.
    1. Line 42: 'medicated' was changed to 'mediated'
    2. Line 43: 'no surprise' was changed to 'not surprising'
    3. Line 146 (now line 150): The extra 'B' in EBNA3 was deleted
    4. Lines 276-278 (now lines 314-318): The sentence was completed and now reads, "Another study seeking to develop EBV-directed therapy is a phase II, prospective, multi-center study assessing the combination of Sintilimab (an anti-PD-1 antibody)  and R-CHOP in previously untreated patients with EBV+ DLBCL, NOS [113]."
    5. Line 284 (now lines 323-324): 'be related to' was deleted. It now reads, "They are thought to originate from...".
    6. Line (now line 336): HRS was defined.
    7. Reference 152 (now reference 177) was updated to the correct reference.
  2. Figure 2 was decreased in size until it fit on one page with its accompanying caption.
  3. Table 1 formatting was adjusted. The width of columns 1 and 4 were decreased, and the spacing after each line is set to the minimum distance. 

Thank you again for the suggestions and please let us know if anything else needs to be addressed. Please see the attachment for the updated manuscript.
